# Impact of Nutritional Status on Postoperative Outcomes in Cancer Patients following Elective Pancreatic Surgery

**DOI:** 10.3390/nu15081958

**Published:** 2023-04-19

**Authors:** Renata Menozzi, Filippo Valoriani, Roberto Ballarin, Luca Alemanno, Martina Vinciguerra, Riccardo Barbieri, Riccardo Cuoghi Costantini, Roberto D’Amico, Pietro Torricelli, Annarita Pecchi

**Affiliations:** 1Division of Metabolic Diseases and Clinical Nutrition, Department of Specialistic Medicines, University Hospital of Modena, 41100 Modena, Italy; 2Hepato-Pancreato-Biliary Surgery and Liver Transplantation Unit, Policlinico Modena Hospital, 41125 Modena, Italy; 3Department of Radiology, University Hospital of Modena, 41224 Modena, Italy; 4Department of Medical and Surgical Sciences of Children and Adults, University Hospital of Modena, 41224 Modena, Italy; 5Unit of Clinical Statistics, University Hospital of Modena, 41224 Modena, Italy

**Keywords:** pancreatic surgery, cancer, nutritional status, reduced muscle mass, sarcopenia, BMI, weight loss, malnutrition

## Abstract

Background: Pancreatic surgery has been associated with important postoperative morbidity, mortality and prolonged length of hospital stay. In pancreatic surgery, the effect of poor preoperative nutritional status and muscle wasting on postsurgery clinical outcomes still remains unclear and controversial. Materials and Methods: A total of 103 consecutive patients with histologically proven carcinoma undergoing elective pancreatic surgery from June 2015 through to July 2020 were included and retrospectively studied. A multidimensional nutritional assessment was performed before elective surgery as required by the local clinical pathway. Clinical and nutritional data were collected in a medical database at diagnosis and after surgery. Results: In the multivariable analysis, body mass index (OR 1.25, 95% CI 1.04–1.59, *p* = 0.039) and weight loss (OR 1.16, 95% CI 1.06–1.29, *p* = 0.004) were associated with Clavien score I–II; weight loss (OR 1.13, 95% CI 1.02–1.27, *p* = 0.027) affected postsurgery morbidity/mortality, and reduced muscle mass was identified as an independent, prognostic factor for postsurgery digestive hemorrhages (OR 0.10, 95% CI 0.01 0.72, *p* = 0.03) and Clavien score I–II (OR 7.43, 95% CI 1.53–44.88, *p* = 0.018). No association was identified between nutritional status parameters before surgery and length of hospital stay, 30 days reintervention, 30 days readmission, pancreatic fistula, biliary fistula, Clavien score III–IV, Clavien score V and delayed gastric emptying. Conclusions: An impaired nutritional status before pancreatic surgery affects many postoperative outcomes. Assessment of nutritional status should be part of routine preoperative procedures in order to achieve early and appropriate nutritional support in pancreatic cancer patients. Further studies are needed to better understand the effect of preoperative nutritional therapy on short-term clinical outcomes in patients undergoing pancreatic elective surgery.

## 1. Introduction

Despite the large number of efforts to enhance the efficacy of varied therapeutic opportunities over the past decade, pancreatic tumors are still one of the most deadly cancers, and five-year survival rates are currently within the range of 6% to 10% [1,2]. Worldwide, exocrine pancreatic cancer is the seventh leading cause of cancer death in both sexes [3].

Among pancreatic cancer patients, survival rates are much better in those who have undergone surgery than those who are unresectable [4]. Regrettably, less than 20% of pancreatic cancer patients will be eligible for resectable surgery [5]. This low resection rate is strongly linked to an advanced cancer stage, the location of the tumor, patients’ comorbidities and an impaired performance status [4]. Indeed, in pancreatic cancer patients, poor oral nutritional intake, catabolism due to malignancy and reduced intestinal absorption because of obstruction or exocrine insufficiency can synergically affect nutritional status and lead to malnutrition and loss of muscle mass [6]. These in turn worsen the patients’ nutritional and performance status and their suitability for surgery.

Pancreatic surgery has been associated with important postoperative morbidity, mortality and prolonged length of hospital stay [7,8,9,10], primarily linked to pancreatic anastomotic leak [11]. Even though technological advances in surgical techniques and perioperative management have greatly improved the mortality rate after pancreatic resection, postoperative morbidity continues to be a significant critical issue [12,13,14].

Pancreatic resection has been identified as one of the most complex surgical procedures as a result of the extended resection, the resulting metabolic stress and the comparatively high rate of complications. This specific kind of surgery strongly modifies metabolic activities and nutritional conditions by triggering inflammation, stress hormones and cytokines. In this specific clinical setting, nutritional status before surgery can also affect postsurgery clinical outcomes. Indeed, in cancer patients, impaired muscle mass before pancreatic surgery is associated with worse long-term survival [15,16,17,18]. However, the effect of preoperative poor nutritional status and muscle wasting on postoperative complications, in-hospital mortality and length of stay still remains unclear and controversial; furthermore, studies on heterogeneity and risk of bias limit the strength of this conclusion [16,17,18,19]. Further research in this area is needed to obtain a definitive answer.

Our retrospective observational study aims to investigate the association between nutritional status before pancreatic elective surgery and short-term clinical outcomes in cancer patients.

## 2. Materials and Methods

This single-center, retrospective study was approved by the local Ethics Committee (n◦: 67/2022/OSS/AOUMO), and all living patients provided written informed consent.

Patients with histologically proven carcinoma undergoing elective pancreatic surgery in University Hospital of Modena from June 2015 through July 2020 were consecutively included and retrospectively studied. No neoadjuvant chemotherapy was administered to enrolled patients. Nutritional assessment was performed before elective surgery as required by local clinical pathway. Oral food intake was assessed by 24 h recall in order to define energy and protein intake. A 24 h dietary recall (24HR) is a structured interview that aims to collect detailed informations and knowledges about all foods, beverages and oral nutritional supplements consumed by patients in the last 24 h. Food models, images and other visual aids were used to support patients in judging and describing volume of portions. Energy requirement was defined in line with European Society for Clinical Nutrition and Metabolism (ESPEN) guidelines on nutrition in cancer patients [20]. Before surgery, computed tomography (CT) scan was performed in order to stage cancer disease and define muscle mass. Diagnosis of cancer-related malnutrition (CRM) was detected in line with Global Leadership Initiative on Malnutrition (GLIM) criteria [21], which include three phenotypic criteria (unintentional weight loss, reduced body mass index and loss of muscle mass) and two etiologic criteria (inflammation and reduced energy intake or absorption). To diagnose malnutrition, at least one phenotypic and one etiologic criterion must be identified [21]. Phenotypic metrics for staging severity of malnutrition as Stage 1 (moderate) and Stage 2 (severe) were available [21].

Clinical and nutritional data were collected in medical records and the hospital electronic medical database at diagnosis and after surgery, including the following variables: age, gender, height, weight, body mass index (BMI), unintentional weight loss %, American Society of Anesthesiology (ASA) score, kind of surgery, operation time and vascular resection.

Postsurgery clinical outcomes (length of hospital stay, morbidity, in-hospital mortality, 30 days reintervention, 30 days readmission, pancreatic fistula, biliary fistula, delayed gastric emptying and digestive hemorrhage) were recorded for all patients. The Clavien score was collected in order to grade adverse events that occur as a result of surgical procedures.

### 2.1. Muscle Mass Measurement

The CT scans performed by the patients for pancreatic disease staging were also used for the evaluation of the body composition, and in particular for the muscle mass analysis. CT investigations were performed with two pieces of equipment: General Electric VCT 64 slice CT scanner (Milwaukee) and General Electric Optima 64 slice CT scanner (Milwaukee). For the reconstructions and the acquisition of the anthropometric parameters, a GE Healthcare AW Volume Share 7 workstation was used, with software that allows one to selectively visualize certain tissues, such as that of muscle, by setting threshold values of density typical of the tissue, and in this case between −29 and +150 Hounsfield units (HU).

Thanks to selective visualization, it was possible to perform a more precise segmentation of the skeletal muscle tissue at the level of the third lumbar vertebra (L3), in which both transverse processes were clearly visible. Areas of interest (ROI) were then drawn using a software tool corresponding to the compartments to be analyzed, within which the area expressed in cm2 and the average density value were calculated automatically.

Total lumbar muscle area (TLA) (cm^2^), including paraspinal and abdominal wall muscles at the L3 level, was calculated by the software after manually tracing an ROI including the psoas muscles, paraspinal muscles (erector spine, quadratus lumborum, multifidus) and wall muscles (transversus, internal and external oblique, rectus abdominis). The skeletal muscle index (SMI) is the parameter obtained from the ratio between the total area of the lumbar muscles (TLA) and the square of the height (cm^2^/m^2^)Figure 1. It is an index of normalization of skeletal muscle mass with respect to the patient’s height. Reduced muscle mass was defined using default sex-specific SMI cutoff values: 52.4 cm^2^/m^2^ for men and 38.5 cm^2^/m^2^ for women [22].

### 2.2. Statistical Analysis

Continuous variables were expressed using mean and standard deviation or median and interquartile range; binary and categorical data were reported as frequencies and percentages.

The associations between length of hospital stay and the patients’ characteristics were assessed using linear regression models, whereas logistic regression models were adopted to investigate the associations with respect to the other outcome measures. In the first place, for each outcome, we performed univariable analyses, and then, when appropriate, a multivariable model was estimated, considering all subjects with nonmissing data. The covariates included in the multivariable models were selected based on the results obtained from the univariable analysis and their clinical importance. In particular, for each outcome, all variables that were statistically associated with that outcome (nominal *p*-value less than 0.05) were selected; furthermore, the main clinical variables of this study, such as the reduced muscle mass indicator, were included. Subsequently, covariates with high association with respect to other covariates were excluded from the models to avoid issues of multicollinearity. Regarding propensity scores, they were used to estimate the probability that a subject has reduced muscle mass, holding other covariates constant. The selection of covariates for the propensity scores was carried out using the same methods described above. A multivariable logistic regression model was then estimated by including in the propensity score model that the variables potentially associated with reduced muscle mass.

Results were reported as the mean difference (MD) or odds ratio (OR) with 95% confidence intervals (CI). Analyses were carried out using R 4.2.1 statistical software.

## 3. Results

### 3.1. Patients’ Characteristics

A total of 103 consecutive patients with a confirmed diagnosis of carcinoma and treated with elective pancreatic surgery in the University Hospital of Modena from June 2015 to July 2020 were retrospectively selected and included in the study. The main characteristics of the enrolled patients are summarized in Table 1. The mean age was 68.7 (±11.2)years, and 59.2% were male. The ASA score was 2 in 61.6% of patients. Pancreatic adenocarcinoma was the most prevalent (76.6%) cancer diagnosis and Vater papilla adenocarcinoma was the second most frequent (10.5%) histological diagnosis. Concerning surgery, pancreaticoduodenectomy was the most common (61.2%) procedure. A vascular resection was performed in 21.2% of surgery. Mean operation time was about 430 min.

Regarding nutritional status before surgery, the mean BMI was 24.6 kg/m^2^ (DS ± 4.6), and unintentional weight loss was detected in 70.5% of the population. An unintentional weight loss higher than 5% was detected in 56.8% of our population. The mean SMI was 43.24 cm^2^/m^2^ (DS ± 13.3), the mean SMI in females was 39.89 cm^2^/m^2^ (DS ± 10.7) and the mean SMI in males was 48.24 cm^2^/m^2^ (DS ± 9.96). A condition of reduced muscle mass was observed in 56.3% of patients. A total of 48% of patients showed an energy oral intake of less than 75% of the daily energy nutritional requirement; differently52% of patients showed an energy oral intake of more than 75% of the daily energy nutritional requirement. Cancer-related malnutrition (CRM) was recognized in 75.7% of patients, in line with GLIM criteria (one phenotypic and one etiologic criterion). Nutritional parameters are summarized in Table 2.

Regarding postsurgery outcomes, the mean length of hospital stay was 16.2 (±11.8) days. We observed a prevalence of 73% (number 73) for postsurgery morbidity and 1.9% (number 2 events) for in-hospital mortality. A Clavien score of I–II was recognized in 53.5% (number 54 events) of patients, and the prevalence of digestive hemorrhage was 10% (number 10 events).

### 3.2. Role of Nutritional Status before Surgery

Our analysis was aimed at searching for clinical and nutritional prognostic parameters. Following adjustment for significantly prognostic covariates at the univariate analysis, a multivariable analysis was performed, which confirmed weight loss before surgery (OR 1.13, 95% CI 1.02–1.27, *p* = 0.027) and total pancreasectomy (OR 8.91, 95% CI 1.79–70.71, *p* = 0.016) as independent prognostic factors in terms of postsurgery morbidity/mortality (Table 3).

We also evaluated the prognostic impact of anthropometric measures before surgery. Following adjustment for significantly prognostic covariates at univariate analysis, a multivariable analysis was performed, which confirmed a significant and independent interaction between BMI (OR 1.25, 95% CI 1.04–1.59, *p* = 0.039), weight loss (OR 1.16, 95% CI 1.06–1.29, *p* = 0.004), reduced muscle mass (OR 7.43, 95% CI 1.53–44.88, *p* = 0.018) and Clavien score I–II. Table 3. Overall, no correlation was highlighted between nutritional status parameters before surgery and length of hospital stay, 30 days reintervention, 30 days readmission and pancreatic fistula (Table 3).

To define the association between reduced muscle mass and each outcome when the outcome’s characteristics did not meet the requirements to estimate the desirable multivariable model, propensity scores were performed to adjust possible differences in covariates. Loss of muscle mass before surgery was associated with postsurgery digestive hemorrhage (OR 0.10, 95%CI 0.01 0.72, *p* = 0.03) (Table 4). No association was identified between reduced muscle mass before surgery, pancreatic fistula, biliary fistula, delayed gastric emptying, Clavien Score III–V, 30 days reintervention and 30 days readmission (Table 4).

## 4. Discussion

Pancreatic resection has been identified as one of the most complex surgical procedures as a result of the extended resection, the resulting metabolic stress and the comparatively high rate of complications. This specific kind of surgery strongly modifies metabolism and nutritional status by triggering inflammation, stress hormones and cytokines [23].

In order to support proper tissue healing and recovery or maintenance of organ functions after surgery, an effective anabolic response and adequate qualitative and quantitative nutritional substrates are required. Malnourished patients deplete their nutritional reserves quickly, which thereby affects their recovery and healing [23]. The development and progression of CRM can be associated with reduced oral nutritional intake and/or increased catabolism [24,25]. Recently, malnutrition has been defined through (one phenotypic and one etiologic criterion) weight loss, low body mass index, muscle wasting, poor energy intake and increased catabolism, in line with GLIM criteria [21].

In our study, the prevalence of CRM before surgery was very high (75.7%); as supposed by some preliminary publications [26,27,28], unintentional weight loss, low BMI, loss of muscle mass and Onodera’s prognostic nutrition index (PNI) have been identified as possible independent prognostic factors for several adverse clinical outcomes after pancreatic surgery.

In particular, BMI (OR 1.25, 95% CI 1.04–1.59, *p* = 0.039) and weight loss (OR 1.16, 95% CI 1.06–1.29, *p* = 0.004) were associated with Clavien score I–II, while weight loss before surgery (OR 1.13, 95% CI 1.02–1.27, *p* = 0.027) affected postsurgery morbidity/mortality.

Our results also highlight the effect of reduced muscle mass before pancreatic surgery on postoperative clinical outcomes, since muscle mass before surgery has been identified as an independent, negative prognostic factor for postsurgery digestive hemorrhages (OR 0.10, 95% CI 0.01 0.72, *p* = 0.03) and Clavien score I–II (OR 7.43, 95% CI 1.53–44.88, *p* = 0.018).

Many publications improperly define sarcopenia only as a condition of reduced muscle mass without performing muscle function measurements as required [29,30,31,32,33,34]. In addition, different tools are available for the assessment of sarcopenia, and the interpretation of results across studies is particularly difficult [16,34]. In pancreatic cancer patients, the impact of preoperative loss of muscle mass on the surgical outcome is still unclear and controversial [16,17,18,19], and available publications show several limitations. In particular, studies included patients receiving pancreatic surgery for both benign and malignant diseases, and not all studies used comparable parameters (different methods, tools and/or cutoffs) to define reduced muscle mass; moreover, a comprehensive nutritional assessment was not performed [16,17,18,19] as required [29,34]. Otherwise, in order to reduce the bias described above, in our study, all included patients had a histologically proven carcinoma, and a global assessment of nutritional status was achieved for each patient before elective pancreatic surgery to diagnose CRM. Furthermore, a quantitative CT analysis of muscle mass was performed by applying the most widely used cutoff for reduced muscle mass as parameter to define malnutrition as recommended [17,19,20,21,22,34]. Indeed, our study was carried out in a high-volume institution for pancreatic surgery, and all the main adverse clinical outcomes after pancreatic surgery were taken into account.

Unfortunately, the retrospective design of this study limits the strength of its conclusions, and for this reason, these findings definitely need to be confirmed in a larger prospective study. Our findings strongly support the relationship between poor nutritional status before pancreatic surgery and short-term adverse clinical outcomes, since not only CT-detected reduced muscle mass but also unintentional weight loss and BMI could negatively affect several short-term clinical outcomes in pancreatic surgery. Further research is needed to better evaluate the effect of severity of malnutrition before surgery on short-term clinical outcomes.

Although they overlap, sarcopenia, reduced muscle mass and CRM are different conditions, the term sarcopenia is unfortunately extensively used to define two different clinical situations: muscle wasting alone and reduced muscle mass associated with an impaired muscle function [29]. This is a significant source of doubts, confusion and mistakes in the research field and in many clinical settings.

Some recent studies have investigated the single effect of the depletion of skeletal muscle mass on short-term clinical outcomes after pancreatic surgery, achieving unclear and controversial results [15,16,17,18,19]. Our findings greatly highlight the need to take not only muscle wasting into account, but all the diagnostic parameters for malnutrition as required by GLIM criteria [21], as part of a nutritional assessment before elective pancreatic surgery.

It should also be remembered that a large number of nutritional assessment tools and scores are available to properly identify cancer patients with malnutrition in surgical settings. Nevertheless, it is still unclear which of these tools are the most appropriate and careful in predicting postoperative adverse outcomes in pancreatic cancer patients [35]. For a long time, hematological biomarkers of status of visceral proteins and liver function have also been used as indicators of impaired nutritional status. Nevertheless, the real predictive efficacy of these biomarkers still remains unclear [36]. Additional research is needed in this area.

Notably, it is strictly recommended that all cancer patients undergoing pancreatic surgery should receive an early, comprehensive and multidimensional evaluation of their nutritional status before elective surgery [23]. Our research supports the advice to assess nutritional status before and after major pancreatic surgery using a validated tool. A multidimensional and comprehensive nutritional assessment is required in order to detect early muscle wasting and/or malnutrition, in line with GLIM criteria, which include three phenotypic criteria (unintentional weight loss, reduced body mass index and loss of muscle mass) and two etiologic criteria (inflammation and reduced energy intake or absorption) [21].

In malnourished patients and in patients at risk of malnutrition, nutritional therapy should be started prior to major cancer surgery, even if operations must be delayed. A period from 7 to 14 days can be suitable [37].

In addition, after elective pancreatic surgery, adjuvant chemotherapy is often indicated to reduce the risk of cancer recurrence. A large number of available publications in this specific clinical setting have reported an impaired response, a reduced tolerance and worse survival rates in pancreatic cancer patients with reduced muscle mass [16]. In light of this, during pancreatic surgery, the early prevention of malnutrition and/or proper perioperative nutritional therapy for CRM is required in order to improve tolerance to antineoplastic therapy and clinical outcome.

In this situation, the introduction of a Nutritional Oncology Board (NOB) in daily practice, aimed at a multidisciplinary assessment of patients and at implementing an early nutritional therapy from oncological diagnosis onward seems to be the right path to take [6]. The NOB, sharing common experiences, goals, obstacles and unmet needs, can be an optimal fertile ground for the birth of collaborative research activities. Indeed, the NOB aims to enhance a shared pathway of care from both a clinical and an organizational point of view, and ideally to also improve awareness towards clinical nutrition [6].

## 5. Conclusions

Our findings highlight that an impaired nutritional status before pancreatic surgery can strongly affect many short-term postoperative outcomes. CRM is a well-known risk factor for surgery-related complications. In cancer patients, before and after pancreatic surgery, proper and appropriate recognition and management of CRM are central clinical concerns and warrant a specific and multidisciplinary (clinical nutrition, oncology, surgery) approach to improve clinical outcomes. The measurement of nutritional status supported by CT analysis of body composition parameters, especially the muscle component, should be a gold standard for preoperative assessment in order to achieve early and appropriate nutritional support.

Further studies are needed to better understand the effect of preoperative nutritional therapy on short-term clinical outcomes in patients undergoing elective pancreatic surgery.

## Figures and Tables

**Figure 1 nutrients-15-01958-f001:**
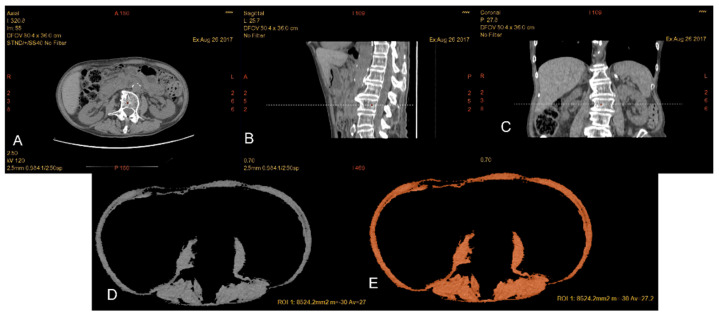
Patient with reduced muscle mass (**A**–**E**): in (**A**), axial CT image at the level of L3 as confirmed by the corresponding reference line in the sagittal (**B**) and coronal (**C**) planes. By applying the threshold −29/+ 150 HU, it is possible to selectively choose the muscle component and draw a ROI including all the musculature at the level of L3 (TLA). Automatically, it is possible to read the overall value of the traced area in cm^2^ and the average value of the density.

**Table 1 nutrients-15-01958-t001:** General characteristics.

		Number	Percentage	Mean	±SD
Gender	Male	61	59.2		
	Female	42	40.8		
Age	Years			68.7	11.2
ASA Score	1	2	2		
	2	59	59.6		
	3	38	38.4		
Site of Cancer	Pancreatic adenocarcinoma	79	76.6		
	NET	6	5.9		
	Vater papilla carcinoma	10	10.5		
	Biliary carcinoma	8	8		
Type of Surgery	Pancreaticoduodenectomy	62	61.2		
	Distal pancreasectomy	6	5.8		
	Total pancreasectomy	35	34		
Operation Time	Minutes			430	107.1
Vascular Resection Performing		21	21.2		

Missing values were excluded from calculations. Abbreviations: ASA: American Society of Anesthesiologists; NET: neuroendocrine tumor; SD: standard deviation.

**Table 2 nutrients-15-01958-t002:** Nutritional parameters before surgery.

		Number	Percentage	Mean	±SD
BMI	kg/m^2^			24.6	4.6
Unintentional Weight Loss %	No weight Loss	28	29.5		
	<5%	13	13.7		
	5–10%	22	23.2		
	≥10%	32	33.6		
Oral Intake	>75%	51	52		
	75–50%	23	23.5		
	<50%	24	24.5		
SMI	Total—cm^2^/m^2^			43.24	13.34
	Female—cm^2^/m^2^			39.89	10.76
	Male—cm^2^/m^2^			48.24	9.96
Reduced Muscle Mass	Yes	58	56.3		
Diagnosis of Malnutrition (GLIM)	Yes	78	75.7		

Missing values were excluded from calculations. Abbreviations: BMI: body mass index; SMI: skeletal muscle index; GLIM: Global Leadership Initiative on Malnutrition.

**Table 3 nutrients-15-01958-t003:** Multivariable analysis for the risk of adverse clinical outcomes.

		Length of Hospital Stay		Morbidity–Mortality		Clavien I–II	
	Category	MD (95% CI)	*p*-Value	OR (95% CI)	*p*-Value	OR (95% CI)	*p*-Value
Age		0.27 (−0.03; 0.56)	0.083	1.01 (0.95; 1.08)	0.694	1.02 (0.96; 1.08)	0.558
Pancreaticoduodenectomy	1	Reference		Reference		Reference	
Distal pancreasectomy	2	−7.88 (−22.16; 6.41)	0.284	0.83 (0.06; 12.54)	0.891	1.33 (0.09; 21.98)	0.830
Total pancreasectomy	3	7.08 (0.83; 13.34)	0.030	8.91 (1.79; 70.71)	0.016	1.07 (0.30; 3.83)	0.913
BMI (kg/m^2^)		−0.14 (−0.84; 0.57)	0.702	1.13 (0.99; 1.35)	0.103	1.25 (1.04; 1.59)	0.039
Unintentional weight loss %		−0.08 (−0.48; 0.32)	0.705	1.13 (1.02; 1.27)	0.027	1.16 (1.06; 1.29)	0.004
ASA	1–2	Reference		Reference		Reference	
ASA	3	1.13 (−5.36; 7.62)	0.734	0.35 (0.07; 1.57)	0.177	0.70 (0.20; 2.43)	0.575
Operation time (min)		0.01 (−0.02; 0.04)	0.682	1.00 (0.99; 1.01)	0.709	0.99 (0.99; 1.00)	0.155
Vascular resection	No	Reference		Reference		Reference	
	Yes	2.61 (−4.43; 9.65)	0.470	0.74 (0.14; 4.24)	0.720	1.35 (0.32; 6.02)	0.685
Reduced muscle mass	No	Reference		Reference		Reference	
	Yes	−2.39 (−10.45; 5.68)	0.564	2.28 (0.37; 18.03)	0.395	7.43 (1.53; 44.88)	0.018

Abbreviations: MD: mean difference; CI: confidence interval; OR: odds ratio; BMI: body mass index; ASA: American Society of Anesthesiologists; min: minutes.

**Table 4 nutrients-15-01958-t004:** Multivariable analysis with propensity scores for the risk of adverse clinical outcomes.

	OR	95% CI		*p*-Value
Pancreatic Fistula				
Reduced muscle mass	1.79	0.35	10.88	0.50
Propensity score	0.28	0.02	2.92	0.30
Biliary Fistula				
Reduced muscle mass	0.41	0.01	18.69	0.62
Propensity score	4.51	0.01	5207.77	0.62
Delayed Gastric Emptying				
Reduced muscle mass	1.78	0.42	8.49	0.45
Propensity score	1.08	0.10	11.21	0.95
Digestive Hemorrhage				
Reduced muscle mass	0.10	0.01	0.72	0.03
Propensity score	101.86	3.07	10,309.36	0.02
Clavien Score III IV				
Reduced muscle mass	0.27	0.06	1.08	0.07
Propensity score	10.24	1.11	119.37	0.05
Clavien Score V				
Reduced muscle mass	0.55	0.01	27.12	0.74
Propensity score	2.13	0.01	1119.19	0.79
30 days Reintervention				
Reduced muscle mass	2.14	0.43	13.45	0.38
Propensity score	2.14	0.16	31.86	0.57
30 days Readmission				
Reduced muscle mass	0.40	0.06	2.75	0.35
Propensity score	1.03	0.05	19.74	0.98

Abbreviations: OR: odds ratio; CI: confidence interval.

## Data Availability

Data are available on request from the authors.

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
