# Peer review of "Impact of Nutritional Status on Postoperative Outcomes in Cancer Patients following Elective Pancreatic Surgery"

_nutrients, 2023, doi:10.3390/nu15081958_

Round 1

Reviewer 1 Report

I appreciate the opportunity to review your manuscript. I think this is an article regarding impact of nutritional status on postoperative outcomes in cancer patients undergoing pancreatic surgery. Overall, most parts of the descriptions are well-described for readers of this Journal but missing in fundamental parts. I was concerned about three points.

Major points

1.     P2Line 77–79. “Post surgery clinical outcomes (length of hospital stay, morbidity, mortality, 30-days re-intervention, 30-days re-admission, pancreatic fistula, biliary fistula, delayed gastric emptying, and digestive hemorrhage)”

Please define the definition of outcomes such as morbidity and mortality (in hospital mortality or 30 days mortality). These detailed definition guaranteed the internal validity of outcomes.

2.     P3Line118–120. Statistical consideration. “In the first place, for each outcome, we performed univariable analyses and then, when appropriate, a multivariable model was estimated, considering all subjects with non-missing data.

Please explain the selection of covariates in statistical methods. And also explain how to build the propensity score models.

3.     P5Line 147–148. ”We observed a prevalence of 73% of post-surgery morbidity. Clavien score I-II was recognized in 53,5% of patients and prevalence of digestive hemorrhage was 10%.”
Please add the each number of outcomes (morbidity, mortality, the events number of Clavien score I–II and digestive hemorrhage).

Author Response

Dr Renata Menozzi

University Hospital of Modena,

Largo del Pozzo 71,

41124, Modena, Italy

[email protected]

Modena, 31/03/2023

Dear Editor,

I am pleased to submit an original research article entitled “IMPACT OF NUTRITIONAL STATUS ON POSTOPERATIVE OUTCOMES IN CANCER PATIENTS FOLLOWING PANCREATIC ELECTIVE SURGERY” for consideration for publication in Nutrients - Special Issue "Nutritional Status and Interventions for Patients with Cancer".

In this manuscript, we have carefully taken into consideration all reviewers comments. All spelling errors and plagiarisms were improved. Following our response to each comment the reviewers made. Any revisions to the manuscript  were marked up using the “Track Changes” function.

1- Reviewer #1 - P2Line 77–79. “Post surgery clinical outcomes (length of hospital stay, morbidity, mortality, 30-days re-intervention, 30-days re-admission, pancreatic fistula, biliary fistula, delayed gastric emptying, and digestive hemorrhage). Please define the definition of outcomes such as morbidity and mortality (in hospital mortality or 30 days mortality). These detailed definition guaranteed the internal validity of outcomes.

Authors’ response: in manuscript mortality outcome was described as required.

  1. Reviewer #1 P3Line118–120. Statistical consideration. “In the first place, for each outcome, we performed univariable analyses and then, when appropriate, a multivariable model was estimated, considering all subjects with non-missing data. Please explain the selection of covariates in statistical methods. And also explain how to build the propensity score models.

Authors’ response: The covariates included in the multivariable models were selected based on the results obtained from the univariable analysis and their clinical importance. In particular, for each outcome, all variables that were statistically associated with that outcome (nominal p-value less than 0.05) were selected; furthermore, the main clinical variables of this study, such as the reduced muscle mass indicator, were included. Subsequently, covariates with high association with respect to other covariates were excluded from the models to avoid issues of multicollinearity. Regarding propensity scores, they were used to estimate the probability that a subject has reduced muscle mass, holding other covariates constant. The selection of covariates for the propensity scores was carried out using the same methods described above. A multivariable logistic regression model was then estimated with reduced muscle mass as the dependent variable, and using this model, the predicted probability of sarcopenia was estimated for each subject.

  1. Reviewer #1 P5Line 147–148. ”We observed a prevalence of 73% of post-surgery morbidity. Clavien score I-II was recognized in 53,5% of patients and prevalence of digestive hemorrhage was 10%.” Please add the each number of outcomes (morbidity, mortality, the events number of Clavien score I–II and digestive hemorrhage).

Authors’ response: In manuscript each number of outcomes was corrected as required.

  1. Reviewer #2 The GLIM criteria are the basis for the definition of malnutrition. As many readers will not be familiar with this classification, it should be described in the methodology

Authors’ response:  GLIM criteria have been described in Material and Method.

  1. Reviewer #2 The authors report that 48% of patients had an oral energy intake below 75% of their nutritional requirements. How was this parameter assessed in a retrospective study?

Authors’ response:  each included patient received a comprehensive  nutritional assessment before elective surgery as required by local clinical pathway. Nutritional assessment also include 24-hour recall in order to define oral food intake (energy kcal/day and protein g/day). Nutritional requirements was define in line with ESPEN guidelines on nutrition in cancer patients. Clin Nutr. 2017 (Bibliography n 20)

  1. Reviewer #2 BMI and weight loss were associated with Clavien Score 1-2. However, the assessement of these uninterventional complications is unreliable according to the literature. Therefore, Clavien Score 3 + 4 should be analyzed separately. Does post-operative morbidity/mortality refer to this question?.

Authors’ response: post-surgery morbidity/mortality refer to this question since the events number of Clavien score III-V and simple size did not allow to performe a multivariable analysis. However to define the association between reduced muscle mass and each outcome when the outcome’s characteristics did not meet the requirements to estimate the desirable multivariable model, propensity scores were performed to adjust possible differences in covariates. No association has been identified between reduced muscle mass before surgery and Clavien Score III-V (table 4).

  1. Reviewer #2 As the consquence of stratifying moderate and severe malnutrition is unclear, a sensitivity analysis of patients with sever malnutrition might be heplful for the interpretation. How many patients had moderate - how many severe malnutrition.

Authors’ response: The assessment of severity of malnutrition was not performed in our study. However this interesting aspect should be consider for further future researches as we mentioned in discussion.

  1. Reviewer #2 The methodology of propensity scores is not provided – were match-pairs created?

Authors’ response: The propensity scores we used were estimated using a multivariable logistic regression model with reduced muscle mass as the dependent variable. Matched pairs were not created for the construction of the propensity scores because it would have substantially reduced the sample size available for the analysis. So, to avoid further statistical issues, the effect of other covariates was adjusted by including in the propensity score model the variables potentially associated with reduced muscle mass.

  1. Reviewer #2 Literature on the effect of nutrition/sarcopenia should be discussed in the discussion (eg Kanada et al. Br J Surg 2011; 98:268). 

Authors’ response: results of study (Br J Surg. 2011 Feb;98(2):268-74) has been discussed. Bibliografy was updated (n 28)

  1. The discussion is too long and lacks a clear focus. Half of it covers the field of sarcopenia, which is not the topic of this paper. Alternatively, sarcopenia should be an item for the mulivariate analysis (and be defined before).

Authors’ response: discussion has been reduced and improved as required.

This manuscript has not been published and is not under consideration for publication elsewhere.

Thank you for your consideration.

Sincerely,

Dr Renata  Menozzi

University Hospital of Modena

Italy

Reviewer 2 Report

This is a well written manuscript, and the analysis covers a very important and timely topic as prehabilitation has become a central issue in oncological surgery. In this analysis the authors detect weight loss and total pancreatectomy as major predictors for morbidity. Subgroup analyses also demonstrate BMI, weight loss and reduced muscle mass as predictors for grade I/II complications.

Criticism:

- The GLIM criteria are the basis for the definition of malnutrition. As many readers will not be familiar with this classification, it should be described in the methodology

- The authors report that 48% of patients had an oral energy intake below 75% of their nutritional requirements. How was this parameter assessed in a retrospective study?

- BMI and weight loss were associated with Clavien Score 1-2. However, the assessement of these uninterventional complications is unreliable according to the literature. Therefore, Clavien Score 3+ 4 should be analyzed separately. Does postoperatiev morbidity/mortality refer to this question?

- As the consquence of stratifying moderate and severe malnutrition is unclear, a sensitivity analysis of patients with sever malnutrition might be heplful for the interpretation. How many patients had moderate - how many severe malnutrition?

- The methodology of propensity scores is not provided – were match-pairs created?

- literature on the effect of nutrition/sarcopenia should be discussed in the discussion (eg Kanada et al. Br J Surg 2011; 98:268). 

- The discussion is too long and lacks a clear focus. Half of it covers the field of sarcopenia, which is not the topic of this paper. Alternatively, sarcopenia should be an item for the mulivariate analysis (and be defined before).

Author Response

(The authors gave the same response as above.)

Round 2
